# Accuracy of Therapeutic Drug Monitoring of Teicoplanin at the Onset of Febrile Neutropenia

**DOI:** 10.3390/medicina59040758

**Published:** 2023-04-13

**Authors:** Masaki Takigawa, Hiroyuki Tanaka, Junichi Suwa, Tomoya Obara, Yohei Maeda, Mamoru Sato, Yoshitomo Shimazaki, Toshihisa Onoda, Akihito Ishigami, Toshihiro Ishii

**Affiliations:** 1Department of Pharmacy, Tokyo Metropolitan Geriatric Hospital, 35-2 Sakae-cho, Itabashi, Tokyo 173-0015, Japan; masaki.taki.79@gmail.com (M.T.);; 2Molecular Regulation of Aging, Tokyo Metropolitan Institute of Gerontology, 35-2 Sakae-cho, Itabashi, Tokyo 173-0015, Japan; 3Department of Practical Pharmacy, Faculty of Pharmaceutical Sciences, Toho University, 2-2-1 Miyama, Funabashi, Chiba 274-8510, Japan; 4Department of Pharmacy, Tokyo Metropolitan Children’s Medical Center, 2-8-28 Musashidai, Huchu, Tokyo 183-8561, Japan

**Keywords:** teicoplanin, febrile neutropenia, hematological malignancy, therapeutic drug monitoring, pharmacokinetics, dosing design

## Abstract

*Background and Objectives*: Teicoplanin (TEIC) is an effective drug for patients with febrile neutropenia (FN); however, it has been reported that these patients may have increased TEIC clearance compared with patients who do not have FN. The purpose of this study was to study therapeutic drug monitoring in patients with FN when the TEIC dosing design was based on the population mean method. *Materials and Methods*: Thirty-nine FN patients with hematological malignancy were included in the study. To calculate the predicted blood concentration of TEIC, we used the two population pharmacokinetic (population PK) parameters (parameters 1 and 2) reported by Nakayama et al. and parameter 3, which is a modification of the population PK of Nakayama et al. We calculated the mean prediction error (ME), an indicator of prediction bias, and the mean absolute prediction error (MAE), an indicator of accuracy. Furthermore, the percentage of predicted TEIC blood concentration within 25% and 50% of the measured TEIC blood concentration was calculated. *Results*: The ME values were −0.54, −0.25, and −0.30 and the MAE values were 2.29, 2.19, and 2.22 for parameters 1, 2, and 3, respectively. For all of the three parameters, the ME values were calculated as minus values, and the predicted concentrations tended to be biased toward smaller values relative to the measured concentrations. Patients with serum creatinine (Scr) < 0.6 mg/dL and neutrophil counts < 100/μL had greater ME and MAE values and a smaller percentage of predicted TEIC blood concentration within 25% of measured TEIC blood concentrations compared with other patients. *Conclusions*: In patients with FN, the accuracy of predicting TEIC blood concentrations was good, with no significant differences between each parameter. However, patients with a Scr < 0.6 mg/dL and a neutrophil count < 100/μL showed slightly inferior prediction accuracy.

## 1. Introduction

Febrile neutropenia (FN) is defined as a fever of more than 38.3 °C or 38 °C for at least 1 h, with a neutrophil count of between 500/µL and 1000/µL or less, with the possibility of falling below 500/µL [1]. FN is a medical emergency requiring the immediate administration of antimicrobial treatment [1]. The first-line antimicrobial agents used for FN are broad-spectrum antimicrobials such as fourth-generation cephems [2,3], carbapenems [4,5], and piperacillin/tazobactam [2]. Glycopeptide antibiotics, such as vancomycin (VCM) and teicoplanin (TEIC), are also used in shock, catheter-related infections, suspected skin and soft tissue infections, and when gram-positive cocci are detected in blood cultures [1].

TEIC is primarily used to treat methicillin-resistant *Staphylococcus aureus* (MRSA) infection [6] and has been reported to have a lower incidence of renal failure than VCM [7]. It has also been reported that TEIC had comparable treatment success rates compared to VCM for gram-positive cocci infections in patients with FN [8]. Hence, TEIC is an effective anti-MRSA agent in patients with FN. Therapeutic drug monitoring (TDM) should be performed when administering TEIC. Area under the curve/minimum inhibitory concentration is reported as a measure of efficacy for TEIC [9]. However, since it is difficult to routinely measure AUC, trough values are measured as an alternative. Target trough concentrations for administering TEIC have been reported to be 15–30 μg/mL [10]. TEIC has pharmacokinetic characteristics such as a high protein binding rate [11] and a very long half-life; therefore, it takes time to reach a steady state [11]. Moreover, TEIC requires an adequate loading dose to achieve target blood concentrations at an early stage. Several methods for the administration of TEIC have been investigated, but there is no established dosing regimen. Furthermore, it has been noted that the clearance of glycopeptide antimicrobials may be increased in patients with cancer and those who have developed FN compared with patients who do not have FN and that standard doses may not achieve the target therapeutic trough values [12,13]. One mechanism of increased clearance due to FN is thought to be augmented renal clearance due to inflammation [14]. To solve this problem, studies have been conducted on the pharmacokinetic parameters specific to patients with cancer [10] and on the optimization of TEIC dosing and blood concentration in patients with hematologic malignancies who develop FN [15,16,17].

One method for designing TEIC dosing is the population mean method using existing population pharmacokinetic (population PK) parameters. In Japan, the population PKs reported by Nakayama et al. [18] are commonly used to predict the blood concentration of TEIC. These population PKs have two parameters, one including albumin (Alb) in the covariates and one excluding Alb. However, only a few reports have examined the predictive accuracy of this TEIC dosing design. Oda et al. [19] reported the possibility of improving the accuracy of predicting TEIC blood concentration by modifying the distribution volume of the population PK parameters in the Nakayama et al. [18] population PKs. Kozono et al. [20] reported on the predictive accuracy of using the estimated glomerular filtration rate (eGFR) calculated using cystatin C to predict the blood concentration of TEIC. In addition, the population PK parameters for TEIC specific to patients with FN have been reported outside of Japan [21]. However, there have been no reports of TEIC blood concentration predictions specifically for patients with FN in Japan.

TEIC is an effective agent for FN, a medical emergency. However, since it has been reported that the clearance of TEIC may be increased in patients with malignancy and FN [12,13], it is meaningful to investigate the accuracy of the population mean method in predicting the blood concentration of TEIC when dosing is designed by this method. Therefore, the purpose of this study was to investigate the accuracy of predicting the blood concentration of TEIC when it was administered by the population mean method to patients with hematological malignancy and FN.

## 2. Materials and Methods

### 2.1. Patients

Patients with hematologic malignancy who were admitted to the Tokyo Metropolitan Geriatric Hospital between June 2013 and March 2021, who were administered TEIC, and whose TEIC blood concentration was measured from days 3 to 5 after the initiation of treatment were included in the study. The following exclusion criteria were used on the eligible patients: patients whose TEIC blood concentration was measured on days other than days 3 to 5 after the start of treatment and patients who did not meet the FN criteria. We collected electronic records of patient data, such as primary disease, age, sex, height, body weight (BW), body mass index (BMI), serum creatinine (Scr), blood urea nitrogen (BUN), total protein (TP), Alb, aspartate aminotransferase (AST), alanine aminotransferase (ALT), and TEIC blood trough concentrations. We also expressed the TEIC dosing status as the total dose on the first and second day of dosing divided by the patient’s BW.

### 2.2. FN Criteria

FN was defined as the following: cases with an axillary temperature of 37.5 °C or higher and a neutrophil count of 500/μL or less or 1000/μL or less and expected to decrease to 500/μL or less within 48 h [22].

### 2.3. Calculation of Predicted TEIC Blood Concentration

To calculate the predicted blood concentration of TEIC, the population PK parameters reported by Nakayama et al. [18] (population PK parameters without Alb (parameter 1) and with Alb as a covariate (parameter 2)), which are widely used in Japan, and the population PK parameters reported by Oda et al. [19] (population PK parameters with a modified distribution volume for parameter 1 (parameter 3)) were used (Table 1). Parameters 1 and 2 are based on data from 120 adult Japanese patients who received TEIC due to MRSA infection. Parameter 3 was created by modifying the distribution volume of parameter 1 to improve prediction accuracy; hence, parameter 3 is similar to parameter 1 except for the distribution volume. Predicted TEIC blood concentrations were calculated from the data of patients in the present study using the population mean method with parameters 1 to 3.

### 2.4. Evaluation of TEIC Blood Concentration Prediction

To evaluate the prediction accuracy of TEIC blood concentration, we calculated the mean prediction error (ME) (Equation (1)) and the mean absolute prediction error (MAE) (Equation (2)).
(1)$$\mathrm{ME}=\Sigma_{i=1}^{n\;}\;(C_{\mathrm{prediction}}-C_{\mathrm{measured}})/n$$
(2)$$\mathrm{MAE}=\Sigma_{i=1}^{n\;}\;|C_{\mathrm{prediction}}-C_{\mathrm{measured}}|/n\;$$

C*_prediction_* and C*_measured_* express the predicted and measured concentrations, respectively. ME is an indicator of prediction bias. For example, a minus value for ME indicates that the C*_prediction_* is biased toward a smaller value than the C*_measured_*. Smaller values of ME also depict a lesser bias. In contrast, MAE is an indicator of accuracy. Smaller values of MAE depict a higher accuracy. A significant bias between the C*_prediction_* and C*_measured_* values was assessed by calculating the 95% confidence interval (CI) of the ME and MAE using the Student’s t-distribution. We also confirmed whether the 95% CI included 0. Furthermore, the percentage of predicted TEIC blood concentrations within 25% and 50% of the actual measured TEIC blood concentrations was calculated.

### 2.5. Statistical Analysis

Continuous variables (age, BW, BMI, TP, Alb, AST, ALT, Scr, BUN, creatinine clearance (Ccr), white blood cell count, neutrophil count, TEIC dosage, and TEIC blood concentration (trough values)) are shown as means (ranges). The Pearson’s correlation coefficient was calculated for the correlation between the predicted and measured blood concentrations of TEIC. Statistically significant differences between the predicted and measured blood concentrations of TEIC were defined as the ME and MAE with the 95% CI differences not containing 0. The Pearson’s correlation coefficient was calculated using SPSS Statistics version 28 (IBM Inc., Armonk, NY, USA), and Microsoft Excel 2016 (Microsoft Corporation, Redmond, WA, USA) was used for all other statistical analyses.

## 3. Results

### 3.1. Patients

A total of 71 patients were included in the investigation, out of which 32 met the exclusion criteria and were excluded and the remaining 39 were included in the study (Figure 1). The background of the target patients is shown in Table 2. The average age of the eligible patients was 65.7 years, with a higher proportion of males compared with females. Many patients were undernourished, with an average Alb concentration of 2.7 g/dL, and had severe neutropenia, with the mean neutrophil count being 108.2/μL. Acute myeloid leukemia was the most common type of hematologic malignancy followed by myelodysplastic syndrome in 22 (56.4%) and 12 (30.8%) patients, respectively.

### 3.2. Correlation between Predicted and Measured TEIC Blood Concentrations

The correlation between predicted and measured TEIC blood concentrations is shown in Figure 2. The correlation coefficients for each population PK parameter were 0.692, 0.690, and 0.713 for parameters 1, 2, and 3, respectively.

### 3.3. Prediction Accuracy When Using Each Parameter

The values of the ME and MAE for each parameter are shown in Table 3. The ME values were −0.54, −0.25, and −0.30 for parameters 1, 2, and 3, respectively and the inclusion of 0 in each 95% CI meant that the respective predicted values were not significantly different from the measured values. Additionally, because the ME is a minus value for all three parameters, the predicted values were smaller than the measured values. The MAE values were 2.29, 2.19, and 2.22 for parameters 1, 2, and 3, respectively, and the respective 95% CIs did not include 0. Therefore, the respective predicted values were significantly different from the measured values. Comparing the ME and MAE values between the BW more than or equal to 50 kg and BW less than 50 kg groups, the ME values tended to be greater in the BW less than 50 kg group. In contrast, the MAE values were comparable in both groups. Similarly, when comparing the BMI less than 18.5 and BMI more than or equal to 18.5 groups, the ME values tended to be greater in the BMI less than 18.5 group, but the MAE values were comparable in both groups. The ME and MAE values in the Alb less than 2.5 g/dL and Alb more than or equal to 2.5 g/dL groups were comparable in parameters 1, 2, and 3, respectively. Comparing the ME and MAE values in the Scr less than 0.6 mg/dL and Scr more than or equal to 0.6 mg/dL groups, the ME and MAE values tended to be greater in the Scr less than 0.6 mg/dL group. When comparing the MAE values for neutrophil count, there was a trend towards smaller MAE values in the 500/μL or more group compared with the less than 100/μL and the 100–500/μL groups. The percentage of predicted TEIC blood concentrations that fall within 25% and 50% of the measured TEIC blood concentrations is shown in Appendix A. For all eligible patients, the proportion falling within 50% exceeded 90% for parameters 1, 2, and 3, respectively. The percentages within 25% were 61.5%, 71.8%, and 64.1% for parameters 1, 2, and 3, respectively, with parameter 2 having the highest percentage. The ME value in the BW less than 50 and BMI less than 18.5 groups tended to be greater than the ME values in the BW more than or equal to 50 kg and the BMI more than or equal to 18.5 groups, but the proportion within the 25th percentile for each parameter generally exceeded 80%. In the Alb less than 2.5 g/dL group, the percentage within 25% was 69.2%, 76.9%, and 61.5% for parameters 1, 2, and 3, respectively, with parameter 2 having the highest percentage. However, in the Scr less than 0.6 mg/dL group, the percentage within 25% was lower for each parameter with 37.5%, 62.5%, and 25.0% for parameters 1, 2, and 3, respectively. In the less than 100/μL neutrophil group, the percentage of patients within 25% was 53.3%, 66.7%, and 60.0% for parameters 1, 2, and 3, respectively, which was lower than in the 100–500/μL and 500/μL or more groups.

## 4. Discussion

In this study, the accuracy of the prediction of TEIC blood concentrations in patients with FN when the dosing design is based on the population mean method using existing population PK parameters was investigated. Although the possibility of increased clearance has been reported in patients with FN, the results of this study suggest that the predictive accuracy of TEIC blood concentrations in patients with FN may be generally good. However, patients with a Scr less than 0.6 mg/dL and a neutrophil count less than 100/μL showed a possible slight decrease in prediction accuracy.

The patient population on which the population PK parameters of Nakayama et al. [12] was based had a mean age of 75.5 (range: 18–96) years, a mean weight of 45.1 (range: 27.0–75.0) kg, a mean Scr of 0.97 (range: 0.23–5.00) mg/dL, and a mean Alb of 2.56 (range: 1.1–4.9) g/dL. The patients in this study are generally considered to be within the range of the population PK patient population of Nakayama et al. [12] despite their younger mean age and larger mean BW. The average neutrophil count of the patients in this study was 108.2/μL, and most patients were severely neutropenic. To the best of our knowledge, no previous reports have examined the accuracy of predicting blood concentrations of TEIC in patients with severe FN in Japan, making this the first report of its kind.

The ME and MAE values in this study are not large considering the intra-individual variability. Therefore, the prediction accuracy of TEIC when using the population mean method is generally good in patients with FN. In the present study, the correlation coefficients between predicted and measured blood concentrations of TEIC were not significantly different for each parameter, although the correlation for parameter 3 was the highest. Similarly, there were no significant differences in the accuracy of predicting TEIC blood levels (ME and MAE) among the parameters. Hence, there were no significant differences between the parameters. Several other studies have also examined the accuracy of predicting TEIC blood concentrations. Kozono et al. [20] investigated the accuracy of predicting TEIC blood concentrations when using parameter 1 in patients with MRSA infection and reported an ME value of −4.24 (95% CI: −6.03–2.44) and an MAE value of 5.14 (95% CI: 3.63–6.66). Oda et al. [19] reported the accuracy of the population mean method using parameters 1, 2, and 3 to predict TEIC blood concentrations in patients with measured TEIC blood concentrations within 48 h of starting treatment. Using parameter 1, they reported an ME value of −3.3 (95% CI: −4.6–−2.1) and an MAE value of 4.0 (95% CI not stated). They also reported that the prediction accuracy using parameter 2 was −3.7 (95% CI: −4.9–−2.5) for the ME and 4.1 (95% CI is not stated) for the MAE. They further reported that parameter 3 had an ME value of 0 (95% CI: −1.1–1.1) and an MAE value of 2.9 (95% CI is not stated), and those whose TEIC blood concentrations were measured within 60 h of starting treatment had an ME value of 0.2 (95% CI: −1.4–1.8) and an MAE value of 3.5 (95% CI is not stated). Compared to their reports, the predictive accuracy in patients with FN included in this study is not inferior. However, this study showed that the prediction accuracy may be slightly reduced in patients with a Scr less than 0.6 mg/dL and neutrophil counts less than 100/μL. In patients with a Scr less than 0.6 mg/dL, this may be due to the Ccr from the Cockcroft–Gault formula, a method of assessing renal function, not accurately representing renal function. A possible solution to this problem may be the use of cystatin C to accurately assess renal function. Recently, the usefulness of the drug dosing design using eGFR calculated from cystatin C has been reported [23,24]. Kozono et al. [20] compared the accuracy of predicting blood concentrations when using Ccr from the Cockcroft–Gault formula from Scr as an assessment of renal function with the eGFR calculated from cystatin C when designing TEIC dosing. They reported that using eGFR calculated from cystatin C did not improve the prediction accuracy. However, in the study by Kozono et al. [20] the median Scr of the included patients was 0.7 (range: 0.5–1.09) mg/dL, and it is possible that there were fewer patients with low creatinine, such as a Scr of less than 0.6 mg/dL. Assessment of renal function using cystatin C may be useful in patients with a Scr of less than 0.6 mg/dL, but further studies are needed. In this study, patients with neutrophil counts less than 100/μL had larger MAE values, while a small percentage of patients had predicted TEIC blood concentrations within 25% of the measured values. Lortholary et al. [21] compared the population PKs of TEIC in patients with hematologic malignancy with severe FN and in healthy adults. They reported that clearance is the main difference in population PK parameters in patients with severe FN compared with population PK parameters in healthy adults. The patients with larger MAE values and the lower proportion of patients with neutrophil counts less than 100/μL whose predicted TEIC blood concentration was within 25% of the measured values observed in this study may be partially due to different clearance rates. This point may be solved by using patient-specific population PK parameters for fever and severe neutropenia as reported by Lortholary et al. [15].

VCM is a glycopeptide like TEIC. An association between FN and serum drug concentration has also been reported for VCM. Hirai et al. [25] reported that ARC associated with FN is associated with low blood levels of VCM. In the present study, it was hypothesized that the accuracy of predicting blood levels of TEIC may be reduced in patients with a Scr less than 0.6 mg/dL and a neutrophil count less than 100/μL. However, it is not clear whether the same is true for VCM. Further studies are needed in this regard.

There are several limitations to this study. First, this is a small, single-center, retrospective study. The small number of cases may have had a substantial impact on a single case. In addition, the patients included in the study may have had underlying diseases, comorbidities, or other patient background biases. Future large-scale studies at multiple centers are needed. Second, the study did not adequately include the effects of the patient’s underlying disease and concomitant medications. We were not able to fully adjust for factors that may affect clearance, such as diuretics, renal failure, and sepsis. In the future, the number of cases should be increased, and a detailed investigation of patient backgrounds should be conducted for further study. Third, this study evaluated the accuracy of predicting TEIC blood concentrations during the first 3 to 5 days after dosing, i.e., the initial loading dose. It did not examine the maintenance dosing period. Therefore, further studies are needed to determine whether the same trend can be obtained in the maintenance period as in the present study.

## 5. Conclusions

In this study, we investigated the accuracy of predicting TEIC blood concentrations in patients with FN using the population mean method of dosage design based on the population PK parameters of Nakayama et al. [18] and Oda et al. [19]. The results suggest that the prediction accuracy of TEIC blood concentrations in patients with FN may be good. Since most of the therapeutic drug monitoring (TDM) analysis software used in clinical practice for TEIC dosing design uses the population PK parameters by Nakayama et al. [18], it is possible that the software can be used for designing dosing in patients with FN. However, patients with a Scr less than 0.6 mg/dL and neutrophil counts less than 100/μL showed a slightly inferior prediction accuracy. The degree of decreased predictive accuracy in these patients is not necessarily significant; however, patients with more than one of these factors may have reduced predictive accuracy and require careful monitoring, including frequent TDM.

## Figures and Tables

**Figure 1 medicina-59-00758-f001:**
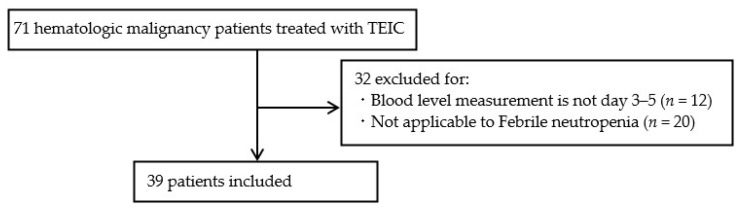
Schema showing the process of screening patients for the study. Abbreviations: TEIC, teicoplanin.

**Figure 2 medicina-59-00758-f002:**
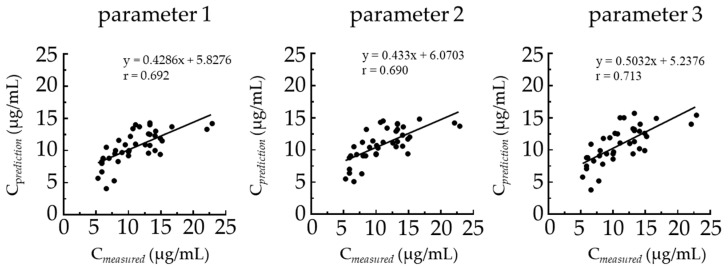
Correlation between the measured and predicted TEIC blood concentrations. Abbreviations: C*_measured_*, measured TEIC blood concentration; C*_predicted_*, predicted TEIC blood concentration; TEIC, teicoplanin.

**Table 1 medicina-59-00758-t001:** Population pharmacokinetic parameters used for the teicoplanin dosage design.

Population Pharmacokinetic Parameter	Parameter 1 (Inter-Individual Variation)	Parameter 2 (Inter-Individual Variation)	Parameter 3 (Inter-Individual Variation)
CL (L/h)	0.00498 × Ccr + 0.00426 × BW (22.1%)	0.0117 × Ccr/Alb + 0.00468 × BW (22.5%)	0.00498 × Ccr + 0.00426 × BW (22.1%)
Vc (L)	10.4 (26.7%)	10.3 (26.1%)	6.6 (26.7%)
K12 (h^−1^)	0.380 (-)	0.384 (-)	0.380 (-)
K21 (h^−1^)	0.0485 (24.5%)	0.0492 (22.6%)	0.0485 (24.5%)

Abbreviations: CL, total clearance; Ccr, creatinine clearance; Alb, albumin; BW, body weight; K12, transfer rate constant from the central compartment to the peripheral compartment. K21; transfer rate constant from the peripheral compartment to the central compartment; Vc, central compartment volume of distribution.

**Table 2 medicina-59-00758-t002:** Baseline characteristics of the patients.

Characteristics	Mean (Range) or No. of Patients (%)
Age (years)	65.7 (31–81)
Sex	
Male	28 (71.8)
Female	11 (28.2)
BW (kg)	56.9 (39.5–90.8)
BMI (kg/m^2^)	21.5 (15.6–35.3)
Total dosage of TEIC for days 1–2 /Body weight (mg/kg)	18.2 (4.4–30.2)
TP (g/dL)	5.9 (3.5–7.9)
Alb (g/dL)	2.7 (1.6–3.8)
AST (U/L)	19.7 (6.0–86.0)
ALT (U/L)	23.5 (7.0–102.0)
Scr (mg/dL)	0.84 (0.46–2.2)
BUN (mg/dL)	19.1 (7.0–68.0)
Ccr (mL/min)	74.9 (24.6–177.8)
White blood cell count (/μL)	1784.1 (0–24,470.0)
Neutrophil count (/μL)	108.2 (0–978.8)
TEIC blood concentration (μg/mL)	11.1 (5.3–22.9)
Hematological malignancy diagnosis	
Acute myeloid leukemia	22 (56.4)
Myelodysplastic syndrome	12 (30.8)
Adult T-cell leukemia/lymphoma	3 (7.7)
Acute lymphocytic leukemia	2 (5.1)

Abbreviations: BW, body weight; BMI, body mass index; TEIC, teicoplanin; TP, total protein; Alb, albumin; AST, aspartate transaminase; ALT, alanine transaminase; Scr, serum creatinine; BUN, blood urea nitrogen; Ccr, creatinine clearance.

**Table 3 medicina-59-00758-t003:** Comparison of prediction accuracy by patient characteristics.

Characteristics		ME (95% CI)			MAE (95% CI)		
		Parameter 1	Parameter 2	Parameter 3	Parameter 1	Parameter 2	Parameter 3
Total (*n* = 39)		−0.54 (−1.51, 0.43)	−0.25 (−1.22, 0.73)	−0.30 (−1.24, 0.64)	2.29 (1.65, 2.93)	2.19 (1.53, 2.85)	2.23 (1.63, 2.82)
Sex	Male (*n* = 28)	−0.37 (−1.51, 0.77)	−0.06 (−1.26, 1.14)	−0.15 (−1.24, 0.94)	2.32 (1.62, 3.02)	2.29 (1.50, 3.08)	2.20 (1.53, 2.86)
	Female (*n* = 11)	−0.96 (−3.13, 1.20)	−0.72 (−2.62, 1.19)	−0.68 (−2.83, 1.47)	2.20 (0.55, 3.85)	1.94 (0.51, 3.36)	2.30 (0.80, 3.80)
Age (years)	<65 (*n* = 9)	−0.38 (−2.54, 1.79)	0.33 (−1.73, 2.39)	−0.18 (−2.42, 2.07)	2.33 (1.26, 3.41)	2.20 (1.15, 3.25)	2.38 (1.23, 3.52)
	≥65 (*n* = 30)	−0.59 (−1.74, 0.57)	−0.42 (−1.58, 0.74)	−0.33 (−1.43, 0.76)	2.27 (1.48, 3.07)	2.19 (1.36, 3.01)	2.18 (1.45, 2.91)
BW (kg)	<50 (*n* = 12)	−1.38 (−3.29, 0.52)	−1.21 (−3.11, 0.69)	−0.98 (−2.88, 0.93)	2.23 (0.72, 3.74)	2.08 (0.54, 3.61)	2.33 (1.04, 3.61)
	≥50 (*n* = 27)	−0.16 (−1.34, 1.01)	0.18 (−0.99, 1.35)	0.004 (−1.13, 1.13)	2.31 (1.59, 3.03)	2.24 (1.49, 2.99)	2.18 (1.47, 2.89)
BMI (kg/m^2^)	<18.5 (*n* = 7)	−1.64 (−4.89, 1.60)	−1.50 (−4.92, 1.92)	−1.26 (−4.28, 1.76)	2.36 (−0.41, 5.13)	2.24 (−0.74, 5.22)	2.29 (−0.06, 4.63)
	≥18.5 (*n* = 32)	−0.30 (−1.33, 0.74)	0.03 (−0.99, 1.05)	−0.09 (−1.10, 0.93)	2.27 (1.64, 2.90)	2.18 (1.55, 2.81)	2.21 (1.60, 2.83)
Alb (g/dL)	<2.5 (*n* = 13)	−0.20 (−2.32, 1.92)	−0.66 (−2.74, 1.41)	0.25 (−1.81, 2.32)	2.48 (1.03, 3.92)	2.20 (0.60, 3.80)	2.64 (1.40,3.88)
	≥2.5 (*n* = 26)	−0.71 (−1.82, 0.41)	−0.04 (−1.17, 1.09)	−0.57 (−1.63, 0.49)	2.19 (1.48, 2.91)	2.18 (1.50, 2.87)	2.02 (1.32, 2.72)
Scr (mg/dL)	<0.6 (*n* = 8)	−1.79 (−5.16, 1.58)	−1.35 (−4.53, 1.83)	−1.65 (−4.92, 1.62)	3.31 (1.04, 5.58)	3.03 (0.96, 5.09)	3.40 (1.49, 5.31)
	≥0.6 (*n* = 31)	−0.22 (−1.19, 0.76)	0.04 (−0.97, 1.05)	0.05 (−0.88, 0.98)	2.02 (1.40, 2.64)	1.97 (1.28, 2.67)	1.92 (1.33, 2.51)
Ccr (mL/min)	<50 (*n* = 8)	−1.63 (−4.31, 1.06)	−1.70 (−4.50, 1.10)	−1.21 (−3.73, 1.31)	2.18 (−0.18, 4.53)	2.23 (−0.26, 4.71)	2.24 (0.36, 4.11)
	≥50 (*n* = 31)	−0.26 (−1.33, 0.81)	0.13 (−0.91, 1.17)	−0.06 (−1.11, 0.99)	2.32 (1.67, 2.96)	2.18 (1.53, 2.83)	2.22 (1.58, 2.87)
Neutrophil count (/μL)	<100 (*n* = 30)	−0.30 (−1.41, 0.81)	0.04 (−1.04, 1.11)	−0.08 (−1.18, 1.01)	2.38 (1.72, 3.04)	2.20 (1.53, 2.88)	2.34 (1.69, 2.98)
	100–500 (*n* = 5)	−2.46 (−7.10, 2.18)	−2.50 (−7.49, 2.49)	−2.22 (−6.09, 1.65)	2.74 (−1.59, 7.07)	2.78 (−1.91, 7.47)	2.30 (−1.48, 6.08)
	≥500 (*n* = 4)	0.08 (−2.14, 2.29)	0.45 (−2.08, 2.98)	0.50 (−2.07, 3.07)	1.03 (−0.15, 2.20)	1.35 (0.37, 2.33)	1.30 (−0.01, 2.61)

Abbreviations: ME, mean prediction error; MAE, mean absolute prediction error; CI, confidence interval; BW, body weight; BMI, body mass index; TP, total protein; Alb, albumin; AST, aspartate transaminase; ALT, alanine transaminase; Scr, serum creatinine; BUN, blood urea nitrogen; Ccr, creatinine clearance.

## Data Availability

Not applicable.

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
