# Peer review of "Accuracy of Therapeutic Drug Monitoring of Teicoplanin at the Onset of Febrile Neutropenia"

_medicina, 2023, doi:10.3390/medicina59040758_

Round 1

Reviewer 1 Report (Previous Reviewer 1)

Dear Author,

Thank you for your response and updating the paper. Your response and update make sense to me. 

Reviewer 2 Report (Previous Reviewer 2)

None, all comments has been addressed 

This manuscript is a resubmission of an earlier submission. The following is a list of the peer review reports and author responses from that submission.

Round 1

Reviewer 1 Report

Authors presented the accuracy of model predicted concentration of teicoplanin in Febrile Neutropenia patients with hematological malignancy. Overall data presentation, discussion, conclusion should be improved to capture the results and implication of the study appropriately. Also there are other published studies outside Japan and that should be discussed too, especially the paper from C.J. Byrne et al, Clinical Microbiology and Infection 23(9) which explored in the same population in more comprehensive way by exploring pk/pd.

Please see the below for the comments.

Abstract

Abstract is described with some vague language. Highly suggested to re-write the abstract to accurately reflect the results

Suggest rewording useful drug. (Introduction section too)

Need to specify Increase CL Compared to which population?

Introduction

Suggest including General pk characteristics in the introduction since that’s one of the main reason for requiring this work. Long half-life, peak/trough ratio/ general variability,etc

Other research outside the us should be discussed

Method

Table 1.Need to elaborate more on the difference between parameter 1 vs paratmeter 3

How many samples were collected for this subjects? Is this enough sample size?

Need more explanation how the Cprediction was derived. Was it calculated from the model by adding the current dataset? Or was it simulated from the model, data independently?

Results

What the impact on the exposure based on the difference in ME/MAE

3.3 Suggest including forest plot to assess the overall effect

Discussion

Need to update the section that repeating the result that was shown in the results section

This could be impacted by the community’s rate of exposed to the previous antibiotics and the resistance. would Bayesian method can be applied and performed better than using simulated value from ppk?

If the C-G formula is not capturing renal function, was MDRD, CKD-EPI tested ? this can be used without cisplatin c data

What’s the benefit/risk profile when use this approach compared to tdm ?

What’s the overall impact of this prediction on this to the dosing.

Conclusion

TDM section need to be explained in the introduction or discussion section.

-          What has been used and what’s the limitation of using tdm if the author proposed to use model predicted value rather than observed value

Suggesting careful monitoring cannot be a conclusion since this is the drug requiring tdm.

Implication of the current assessment need to be reflected here

Reviewer 2 Report

Overall, good study with strong results, however, I have more edits to have better presentation, my comments are as follows”

1-    Title: this statement of the title is vague please consider rephrasing “ at the Onset of Febrile Neutropenia”

2-    Title: I suggest changing predicting blood concentration with “therapeutic drug monitoring because this is the correct phrase used in the literature

3-    Abstract” change “Background and objectives” to “Background”

Abstract: rephrase the background it is a bit confusing in this sentence “ when the dosing design of TEIC was performed using the population mean method.”

4-    Abstract: very long you exceeded 250 words based on journal format

5-    Introduction is very long, please try to shorten and remove excess details

6-    PPK abbreviation should be replaced with population PK that is more consistent with the literature

7-    Overall, good presentation of the results, one point however, could you move Table 4 to the supplementary section.? I feel too much to include in the manuscript

8-    Can you add a paragraph in the discussion section about vancomycin? Since teicoplanin and vancomycin are both glycopeptides, could vancomycin also affect clearance in patients with FN? We need to have info about that in the manuscript 

9-    Add space after the conclusion section to separate author contribution

10- In the introduction, briefly add what are the possible mechanisms for increasing the clearance of teicoplanin in patients with FN